# Important Role of Endogenous Nerve Growth Factor Receptor in the Pathogenesis of Hypoxia-Induced Pulmonary Hypertension in Mice

**DOI:** 10.3390/ijms24031868

**Published:** 2023-01-18

**Authors:** Chiaki Goten, Soichiro Usui, Shin-ichiro Takashima, Oto Inoue, Kosei Yamaguchi, Daiki Hashimuko, Yusuke Takeda, Ayano Nomura, Kenji Sakata, Shuichi Kaneko, Masayuki Takamura

**Affiliations:** 1Department of Cardiovascular Medicine, Graduate School of Medical Science, Kanazawa University, 13-1 Takara-machi, Kanazawa 920-8641, Ishikawa, Japan; 2Department of Information-Based Medicine Development, Graduate School of Medical Science, Kanazawa University, 13-1 Takara-machi, Kanazawa 920-8641, Ishikawa, Japan

**Keywords:** nerve growth factor receptor (Ngfr), pulmonary arterial hypertension (PAH), vascular remodeling

## Abstract

Pulmonary arterial hypertension (PAH) remains a disease with poor prognosis; thus, a new mechanism for PAH treatment is necessary. Circulating nerve growth factor receptor (Ngfr)-positive cells in peripheral blood mononuclear cells are associated with disease severity and the prognosis of PAH patients; however, the role of Ngfr in PAH is unknown. In this study, we evaluated the function of Ngfr using Ngfr gene-deletion (Ngfr^−/−^) mice. To elucidate the role of Ngfr in pulmonary hypertension (PH), we used Ngfr^−/−^ mice that were exposed to chronic hypoxic conditions (10% O_2_) for 3 weeks. The development of hypoxia-induced PH was accelerated in Ngfr^−/−^ mice compared to littermate controls. In contrast, the reconstitution of bone marrow (BM) in Ngfr^−/−^ mice transplanted with wild-type BM cells improved PH. Notably, the exacerbation of PH in Ngfr^−/−^ mice was accompanied by the upregulation of pulmonary vascular remodeling-related genes in lung tissue. In a hypoxia-induced PH model, Ngfr gene deletion resulted in PH exacerbation. This suggests that Ngfr may be a key molecule involved in the pathogenesis of PAH.

## 1. Introduction

Pulmonary arterial hypertension (PAH) is a disease with poor prognosis that causes progressive intimal hyperplasia and medial thickening of the pulmonary artery (PA), thus increasing pulmonary vascular resistance and resulting in right heart failure [1]. Pulmonary hypertension (PH) is characterized by structural and functional changes in the pulmonary arterial vasculature that originate in increased pulmonary vascular resistance and progress from isolated intimal thickening to plexiform fibrosis in the terminal stages [2]. Pulmonary vascular remodeling is an important component of the etiology of PH. Bone marrow (BM)-derived progenitor cells and perivascular inflammatory infiltrates contribute to the process of PA remodeling [3]. Several studies have suggested that the process of vascular remodeling is partially driven by BM-derived angiogenic cells, with endothelial progenitor cells and mesenchymal stem cells (MSCs) contributing to pulmonary vascular remodeling [4,5].

MSCs are pluripotent stem cells that primarily reside in BM and have the potential to differentiate into mesenchymal tissue lineages. They inhibit inflammatory sites and secrete a variety of soluble factors including growth factors, cytokines, and chemokines. They are also involved in the repair of damaged tissue by secreting paracrine factors. In addition, circulating mesenchymal progenitor cells have been shown to accumulate in the walls of remodeled vessels of animal models and PAH patients, thereby contributing to vessel wall thickening [6].

Nerve growth factor receptor (Ngfr) plays an important role in nerve cell growth, development, and differentiation, especially during the embryonic period [7]. Its expression decreases after birth; however, once tissue damage occurs, its expression increases in an attempt to repair the damaged tissue [8]. Also known as CD271, Ngfr has been implicated in MSC differentiation and the enrichment of several progenitor/stem-cell subtypes. Its expression in peripheral blood mononuclear cells (PBMCs) is associated with the progression of vascular remodeling [9]. Additionally, Ngfr-positive cells in PBMCs have been linked to the severity and prognosis of PAH [10]. These results suggest that circulating Ngfr-positive cells in PBMCs may be involved in the pathophysiology of PAH; however, their specific role remains unclear. In this study, we examined the effects of hypoxia on PH using Ngfr gene-deficient mice.

## 2. Results

### 2.1. Frequency of Ngfr-Positive Cells

The circulating Ngfr-positive cells were present in a detectable range in mouse PBMCs (Figure 1A–C) and their frequency was increased in the PBMCs of WT mice under hypoxic conditions compared to that in WT mice under normoxic conditions (Figure 1D). The gene expression of Ngfr was significantly increased in the lung under hypoxic conditions compared to that under normoxic conditions (Figure 1E).

To assess the localization of Ngfr-positive cells in lung tissue, immunohistological analysis was performed on WT mice under hypoxic conditions. Ngfr-positive cells were detected in the alveolar tissue but not in vascular smooth muscle cells (Figure 1F,G).

### 2.2. Deficiency of the Ngfr Gene Enhances Hypoxia-Induced Pulmonary Hypertension

Right cardiac catheterization was performed to accurately assess hemodynamic parameters and to confirm whether the hypoxia-induced PH model was sufficiently created in the mice. Under normoxic conditions, there were no differences between Ngfr^−/−^ and WT mice regarding RAP or RVSP (Figure 2A,B). Hypoxia resulted in an increase in RVSP in WT mice. Interestingly, the elevation in RAP and RVSP under hypoxic conditions was significantly increased in Ngfr^−/−^ mice compared to that in WT mice (Figure 2C,D). Furthermore, the hypoxia-induced increase in the Fulton’s ratio was significantly enhanced in Ngfr^−/−^ mice compared to that in WT mice (Figure 2E).

### 2.3. Deficiency of the Ngfr Gene Enhances Hypoxia-Induced Pulmonary Vascular Remodeling

Immunohistochemical analysis was performed to assess the pulmonary arterial remodeling observed in PAH. Under normoxic conditions, PA morphology was similar between Ngfr^−/−^ and WT mice (Figure 3A,B). After 3 weeks of hypoxia, Ngfr^−/−^ mice exhibited increased muscularization of the distal PAs compared to WT mice. The hypoxia-induced increase in fibrosis around the PAs was significantly enhanced in Ngfr^−/−^ mice compared to that in WT mice (Figure 3C–E).

### 2.4. BM-Specific Depletion of Ngfr Exacerbates Hypoxia-Induced Pulmonary Hypertension

To confirm the accumulation of BM-derived cells in the lung, wild-type mice were lethally irradiated, transplanted with BM cells from green fluorescent protein (GFP) transgenic mice, and bred under hypoxic conditions. After 3 weeks of hypoxia, FCM analysis revealed that the frequency of Ngfr-positive cells was significantly increased by hypoxic exposure compared to control conditions (Figure 4A,B). Furthermore, immunohistochemistry analysis revealed that frequencies of Ngfr-positive and GFP-positive cells in the lung were increased under hypoxic conditions compared with those under control conditions (Figure 4C). To examine whether BM-derived Ngfr-positive cells were involved in the development of PH, wild-type mice were lethally irradiated and transplanted with BM cells from Ngfr^−/−^ mice. After 3 weeks of hypoxia, RAP and RVSP were significantly increased in BM-specific Ngfr^−/−^ mice compared to WT mice under hypoxic conditions. In contrast, RAP and RVSP were significantly reduced in Ngfr^−/−^ mice transplanted with BM cells from WT mice, compared to Ngfr^−/−^ mice (Figure 4D,E).

### 2.5. Deficiency of Ngfr Increases Inflammatory Cytokines in the Hypoxic Lung

qPCR was performed to assess the changes in biological markers leading to pulmonary arterial remodeling in the lung tissues of WT and Ngfr^−/−^ mice under hypoxic conditions. The mRNA expression of BMPR2 was significantly reduced in the lung tissue of Ngfr^−/−^ mice compared to that in WT mice under hypoxic conditions (Figure 5A). On the other hand, mRNA expression levels of Tgfb1, Pai-1, Il-6, and Tnf were significantly increased in the lung tissue of Ngfr^−/−^ mice compared to those in WT mice under hypoxic conditions (Figure 5B–E).

## 3. Discussion

Our results suggest that endogenous Ngfr plays a key role in preventing the development of PH. The accumulation of Ngfr-positive cells was increased in response to hypoxia, and hypoxia-induced PH was significantly enhanced in Ngfr^−/−^ mice. Furthermore, BM-specific depletion of Ngfr promoted PH in response to hypoxia. We propose that Ngfr-positive cells have protective effects against vascular remodeling in a hypoxia-induced mouse model of PH.

Ngfr, also known as p75NTR, is a transmembrane receptor that mediates cell survival, as well as the death of neural cells. The absence of Ngfr leads to a severe phenotype, including defects in the vascular system [11]. Previously, our laboratory used circulating Ngfr-positive cells as an indicator of disease severity in patients with idiopathic PAH and PAH associated with scleroderma [10]. However, the role of Ngfr-positive cells was unknown; thus, we examined the mechanisms in a mouse model of hypoxia exposure in this study. In support of previous results, we found that Ngfr-positive cells showed increased numbers in peripheral blood and accumulated in the lungs under hypoxic conditions. To examine the role of Ngfr in the pathogenesis of PH, we investigated a hypoxia-induced PH model in Ngfr^−/−^ mice. PH responses, as assessed by the measurement of RV hypertrophy and RVSP, were higher in Ngfr^−/−^ mice. The augmented pulmonary hypertensive response in Ngfr^−/−^ mice strongly implicated Ngfr in the pathogenesis of PH.

BM-derived cells contribute to vascular remodeling in experimental PH [12,13]. In our study, to track BM-derived cells in vivo, we generated GFP BM chimeric mice and subsequently subjected them to hypoxia-induced PH. We showed the presence of BM-derived GFP-positive cells in the remodeled PAs. It is likely that the Ngfr-positive cells migrated into the pulmonary arterial regions from BM. Here, we have demonstrated that BM-derived Ngfr-positive cells played an important role in pathological pulmonary arterial remodeling in the hypoxia-induced PH model. Hypoxia-induced PH was augmented in WT mice transplanted with Ngfr^−/−^ BM cells, as well as in Ngfr^−/−^ mice. Furthermore, hypoxia-induced PH was ameliorated in Ngfr^−/−^ mice transplanted with WT BM cells. These results suggest that Ngfr-positive cells play a role in protection against hypoxia-induced PH. We speculate that Ngfr-positive cells migrate to the lung tissue and secrete factors that inhibit PH progression in a paracrine manner. BMT of Ngfr-overexpressing cells or replacement therapy with protective factors secreted by Ngfr-positive cells may partially suppress PH progression.

Freund-Michel et al. reported that the administration of a neutralizing antibody to NGF, a low-affinity ligand of NGFR, suppressed hypoxia-induced PH in rats [14]. The authors showed that blockade of the NGF/NGFR axis suppressed the production of inflammatory cytokines in vascular smooth muscle cells and proposed that NGF may mediate PH. These results are in contrast with our findings that the genetic deletion of NGFR promotes PH. This difference may be due to the fact that Freund-Michel et al. administered a neutralizing antibody for NGF to the hypoxia model rats, whereas we used a genetic deletion model of NGFR. Signaling through NGFR has different biological effects depending on where it can heterodimerize with other types of receptors, including tropomyosine receptor kinases (Trks) A, B, and C; neurotensin receptor 1 and 2; and sortilin [15]. The co-expression of NGFR and Trks promotes cell survival, while the co-expression of NGFR and sortilin promotes cell death. Elucidating the effects of genetic deletion of NGFR on the activation of other types of receptors requires further investigation.

Anti-inflammatory effects on PAs have been shown to prevent disease development in PH models. Loss or dysfunction of the balance between TGF-β1 and BMPR2 plays a critical role in PAH predisposition and disease progression [16]. A loss of BMPR2 favors the endothelial-to-mesenchymal transition, allowing cells of myofibroblastic character to create a vicious feed-forward process that leads to hyperactivated TGF-β signaling [17]. This study also found that Ngfr gene-deletion mice showed lower BMPR2 expression and increased Tgfb1 gene expression in response to hypoxic stimulation. Taken together, these results suggest that Ngfr may play a key role in maintaining BMPR2 and TGFb1 homeostasis. Additionally, Ngfr deletion may lead to an imbalance between Tgfb1 and BMPR2, thereby contributing to the exacerbation of PAH.

PAI-1 is a member of the serpin (serine proteinase inhibitor) family and acts as an important inhibitor of fibrinolysis and modulator of cellular responses linked to vascular remodeling [18]. The serum levels of tPAI-1 may be useful for predicting severity, similar to mPAP in patients with PAH [19,20]. These reports suggest that PAI-1 is involved in pulmonary vascular remodeling and the pathogenesis of PH, supporting elevated PAI-1 expression and exacerbation of PAH in this study.

Elevated levels of the proinflammatory cytokine IL-6 have been reported in PAH [21]. The increased IL-6 production in PAH is thought to reflect enhanced synthesis by both inflammatory and pulmonary vascular cells [22]. IL-21 promotes PAH in association with M2 macrophage polarization downstream of IL-6-signaling [23]. These reports support the importance of pro-inflammatory cytokines in PAH and suggest that the pro-inflammatory effect of Ngfr gene deletion may result in PAH exacerbation. The TNFα-mediated suppression of BMPR-II subverts BMP signaling, leading to BMP6-mediated PASMC proliferation via the preferential activation of the ALK2/ACTR-IIA signaling axis. Furthermore, TNFα, via SRC family kinases, increases pro-proliferative NOTCH2 signaling in HPAH PASMCs with reduced BMPR-II expression [24]. These results support the exacerbation of PAH associated with altered genetic profiles due to Ngfr gene deletion.

This study had several limitations. First, hypoxia-induced PH model mice, although commonly used, do not accurately reflect the pathology of PAH patients. Second, Ngfr-deletion PH model mice altered the gene profile that affected pulmonary vascular remodeling and PAH exacerbation. Based on the experimental results of bone marrow transplantation, it is speculated that BM-derived NGFR-positive cells migrate to local lung lesions and secrete protective factors in a paracrine manner, but the detailed mechanisms remain unclear. Thus, further studies are needed to evaluate the mechanisms of Ngfr-associated vascular remodeling in PAH. Third, this study was not able to assess the interactions between Ngfr and other primary co-expressed receptors; thus, future single-cell-based analysis is also required.

In summary, our results show that circulating Ngfr-positive cells suppressed vascular remodeling and improved hemodynamic parameters, such as RVSP and RAP, in hypoxia-induced PH model mice. The gene deletion of Ngfr altered the gene profile associated with pulmonary vascular remodeling in lung tissue, which may lead to PAH exacerbation.

## 4. Materials and Methods

### 4.1. Ethics Statement

All animal studies were performed according to the Guide for the Care and Use of Laboratory Animals at Kanazawa University, which strictly conforms to the Guide for the Care and Use of Laboratory Animals published by the United States National Institutes of Health (NIH, Bethesda, MD, USA). The protocol was approved by the ethics committee of Kanazawa University (Approval NO AP-173898, 19 April 2019).

### 4.2. Animal Models and Experimental Procedures

Ngfr gene-deletion (Ngfr^−/−^) mice carrying a mutation of the gene encoding Ngfr by targeted mutation in embryonic stem cells were obtained from the Jackson Laboratory (Bar Harbor, ME, USA). Adult 8-week-old Ngfr^−/−^ mice (*n* = 12) and littermate wild-type mice (*n* = 12) were placed in a ventilated chamber and exposed to 10% O_2_ (hypoxic conditions) or room air (normoxic conditions) for 3 weeks, respectively (Ngfr^−/−^ normoxia, *n* = 6; Ngfr^−/−^ hypoxia, *n* = 6; WT normoxia, *n* = 6; WT hypoxia, *n* = 6). All mice were fed a standard diet of rodent chow and housed in a controlled environment (temperature, 23 ± 2 °C; humidity, 55 ± 10%; 12 h light/dark cycle). After 3 weeks, the mice were anesthetized with isoflurane, and right catheterization was performed to evaluate the right atrial and right ventricular systolic pressures (RAP and RVSP, respectively). Blood samples were collected for flow cytometry (FCM) analysis. After sufficient reflux with phosphate-buffered saline (PBS), the lung was excised, fixed in formalin, and cryopreserved for immunohistochemical analysis. The heart was also excised for measurement of the weight of the right ventricle (RV), left ventricle (LV), and septum (S) in order to calculate Fulton’s index, with Fulton’s index = RV/(LV + S).

### 4.3. BM Transplantation

BM transplantation was performed using standard methods [25]. BM cells were collected from the femur and tibia of donor WT, green fluorescent protein (GFP) transgenic (GFP-Tg), and Ngfr^−/−^ mice between 8 and 12 weeks. The recipient mice were irradiated at a total of 9.6 Gy. Then, the irradiated mice were transplanted with BM 24 h later and housed in room air for 4 weeks to allow for complete replacement with transplant-derived peripheral blood (WT transplanted with GFP-Tg BM under normoxic conditions, *n* = 6; WT transplanted with GFP-Tg BM under hypoxic conditions, *n* = 6; WT transplanted with Ngfr^−/−^ BM under hypoxic conditions, *n* = 6; Ngfr^−/−^ transplanted with WT BM under hypoxic conditions, *n* = 6).

### 4.4. Hemodynamic Measurements and Morphometric Analysis

After anesthesia with isoflurane (induction 4%, maintenance 1% mixed room air), the right jugular vein was exposed, and an SPR-671 pressure transducer catheter (Micro-Tip 1.4F, Millar Instruments) was inserted into the jugular vein via a small incision and passed into the right ventricle. Data acquisition was performed using the Powerlab data system (AD Instruments, Sydney, Australia). Blood samples were collected for FCM analysis. After sufficient reflux with PBS, the lung was excised, fixed in formalin, and cryopreserved. The heart was also excised, the atria were removed, and the RV wall was separated from the LV and septum. The weight of each ventricle was measured. RV hypertrophy was determined by Fulton’s ratio/index [RV/(LV + S)].

### 4.5. Histological Analysis

The lung specimens were fixed in 4% paraformaldehyde, embedded in paraffin, and cut into 4 μm-thick sections for staining, as described previously [26]. For immunohistochemical staining, the specimens were stained with anti-Ngfr antibody (AB1554, Sigma-Aldrich, Japan). For immunofluorescence, the specimens were stained with anti-smooth muscle actin antibody (Abcam, Cambridge, UK) with Alexa 488-conjugated secondary antibodies and anti-Ngfr antibody with Alexa 594-conjugated secondary antibodies (Life Technologies Japan, Tokyo, Japan). The nuclei were also stained with 4′,6-diamidino-2-phenylindole (DAPI; Vector Laboratories Inc., Burlingame, CA, USA). The ratio of pulmonary arterial medial thickness to total vessel size (media/cross-sectional area [CSA]) was calculated using Image J software (NIH). Images were acquired using a microscope (BZ-X800; Keyence, Osaka, Japan).

### 4.6. Flow Cytometry

A 0.5 mL blood sample was collected in a heparinized tube, and PBMCs were isolated using Ficoll-PaqueTM (Miltenyi Biotec, Tokyo, Japan). Lung pieces were digested in Dulbecco’s Modified Eagle Medium containing Liberase (40 μg/mL, Roche, Mannheim, Germany) using a gentleMACS dissociator (Miltenyi Biote, Tokyo, Japan). The cells were filtered through a 100 μm cell strainer and washed twice with MACS buffer (PBS, 0.5% bovine serum albumin (BSA), 2 mmol/L of ethylenediaminetetraacetic acid, degassed). Then, the cells were fixed using lysing buffer (1×) (BD Biosciences, San Jose, CA, USA) and incubated for 10 min in the dark. The cells were centrifuged at 400× *g* for 5 min at 4 °C and washed with MACS buffer. After re-centrifugation, the cells were resuspended with MACS buffer, to which 100 μL was added to each fluorescence-activated cell sorting tube. Then, the cells were incubated in 5 μL of Mouse BD Fc block (Purified Rat Anti-Mouse CD16/CD32, BD Bioscience, USA) for 10 min at 4 °C in the dark. Immunophenotyping was conducted by staining for 15 min at 4 °C in the dark with fluorochrome-labeled monoclonal antibodies LNGFR-APC (human and mouse 130-116-497, BD Bioscience). MACS buffer (1000 μL) was added to the samples, which were then centrifuged at 400× *g* for 5 min at 4 °C. The samples were resuspended in 150 μL of MACS buffer for FCM analysis. Untreated cells served as the negative control to compensate for the fluorochrome overlap. The fluorescence intensity of the cells labeled with the fluorochrome-labeled monoclonal antibodies was examined using an Accuri C6 flow cytometer (BD Bioscience). The FCM analysis was performed using FlowJo software version 10 (BD Bioscience).

### 4.7. Quantitative Real-Time Polymerase Chain Reaction (qPCR)

qRT-PCR was performed as previously described [27]. Total RNA was isolated using ISOSPIN Cell & Tissue RNA (Nippon Gene, Tokyo, Japan). Then, 100 ng of total RNA was transcribed to cDNA using the TaqMan Universal Master Mix. Mouse beta-actin (4352341E, Applied Biosystems, Foster City, CA, USA) was used as the endogenous control. qPCR was performed using a QuantStudio 12K Flex instrument (Applied Biosystems). The primers and probes were obtained from Thermo Fisher Scientific (Waltham, MA, USA). The assay identification of each primer and probe are as follows: bone morphogenetic protein receptor type-2 (BMPR2; Mm03023976_m1); transforming growth factor-b1 (Tgf-b1; Mm 01178820_m1); plasminogen activator inhibitor-1 (PAI-1; Mm00435860_m1); interleukin-6 (IL-6; Mm00446190_m1); and tumor necrosis factor (TNF; Mm00443258_m1).

### 4.8. Statistical Analysis

The statistical analysis was performed using Graph Pad Prism 9.0 software (Graph Pad Software, La Jolla, CA, USA). An unpaired t-test was used to compare two groups. For comparisons between four or more groups, one-way analysis of variance was conducted, followed by a post hoc Bonferroni–Dunn’s comparison test to determine the statistical significance of differences between the mean values. Statistical significance was defined as a *p*-value < 0.05.

## Figures and Tables

**Figure 1 ijms-24-01868-f001:**
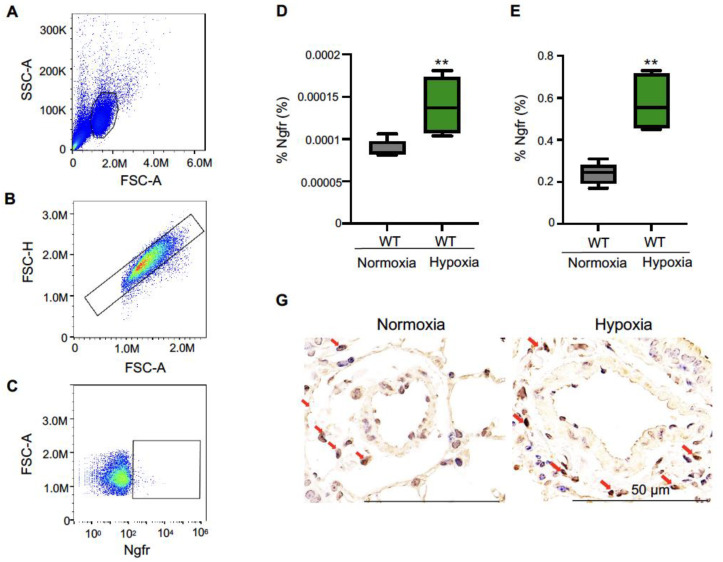
Increase in frequency of Ngfr-positive cells in WT mice under hypoxic conditions. FCM was performed to assess the frequency of Ngfr-positive cells in PBMCs of WT mice under normoxic and hypoxic conditions. Representative flow cytometric figures (mononuclear cells (**A**), single cells (**B**), and Ngfr-positive cells (**C**)). (**D**) The frequencies of Ngfr-positive cells in PBMCs (% Ngfr) of WT mice under normoxic and hypoxic conditions. (**E**) Ngfr expression in mouse lung in response to hypoxia. (**F**) Immunofluorescence for Ngfr in lung under normoxic and hypoxic conditions. (**G**) Immunostaining for Ngfr in lung under normoxic and hypoxic conditions. Red arrows show Ngfr-positive cells. FCM, flow cytometry; SSC, side scatter; FSC, forward scatter; Ngfr, nerve growth factor receptor; PBMC, peripheral blood mononuclear cell; WT, wild type. Center lines show the medians, box limits indicate the 25th and 75th percentiles, whiskers extend 1.5× the interquartile range from the 25th and 75th percentiles. ** *p* < 0.01 versus WT Normoxia.

**Figure 2 ijms-24-01868-f002:**
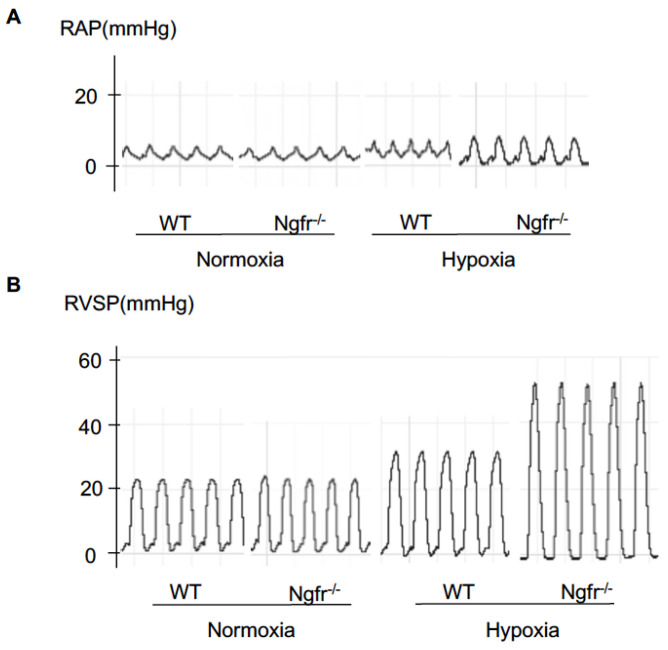
Hemodynamic parameters (RAP and RVSP) in WT and Ngfr^−/−^ mice under normoxic and hypoxic conditions. (**A**) The waveforms of RAP in WT and Ngfr^−/−^ mice under normoxic and hypoxic conditions. (**B**) The waveforms of RVSP in WT and Ngfr^−/−^ mice under normoxic and hypoxic conditions. (**C**) RAP of WT and Ngfr^−/−^ mice under normoxic and hypoxic conditions. (**D**) RVSP of WT and Ngfr^−/−^ mice under normoxic and hypoxic conditions. (**E**) Fulton’s ratio (RV/LV+S) of WT and Ngfr^−/−^ mice under normoxic and hypoxic conditions. RAP, right atrial pressure; RVSP, right ventricular systolic pressure; RV, right ventricle; LV, left ventricle; S, septum. Center lines show the medians, box limits indicate the 25th and 75th percentiles, whiskers extend 1.5× the interquartile range from the 25th and 75th percentiles. ** *p* < 0.01 versus WT Normoxia. ^##^
*p* < 0.01 versus Ngfr^−/−^ Normoxia. ^∫∫^
*p* < 0.01 versus WT Hypoxia.

**Figure 3 ijms-24-01868-f003:**
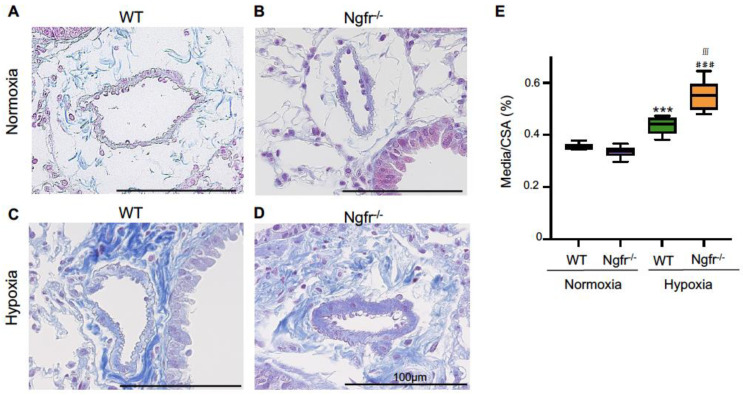
Histological analysis in pulmonary arterioles of WT and Ngfr^−/−^ mice under normoxic and hypoxic conditions. Azan staining was performed to assess pulmonary vascular remodeling in WT and Ngfr^−/−^ mice under hypoxic conditions. Azan staining of lung tissue in (**A**) WT Normoxia, (**B**) Ngfr^−/−^ Normoxia, (**C**) WT Hypoxia, and (**D**) Ngfr^−/−^ Hypoxia. (**E**) The ratio of the media to CSA (media/CSA) of WT and Ngfr^−/−^ mice under normoxic and hypoxic conditions. CSA, cross-sectional area. Center lines show the medians, box limits indicate the 25th and 75th percentiles, whiskers extend 1.5× the interquartile range from the 25th and 75th percentiles. *** *p* <0.001 versus WT Normoxia. ^###^
*p* < 0.001 versus Ngfr^−/−^ Normoxia. ^∫∫∫^
*p* < 0.001 versus WT Hypoxia.

**Figure 4 ijms-24-01868-f004:**
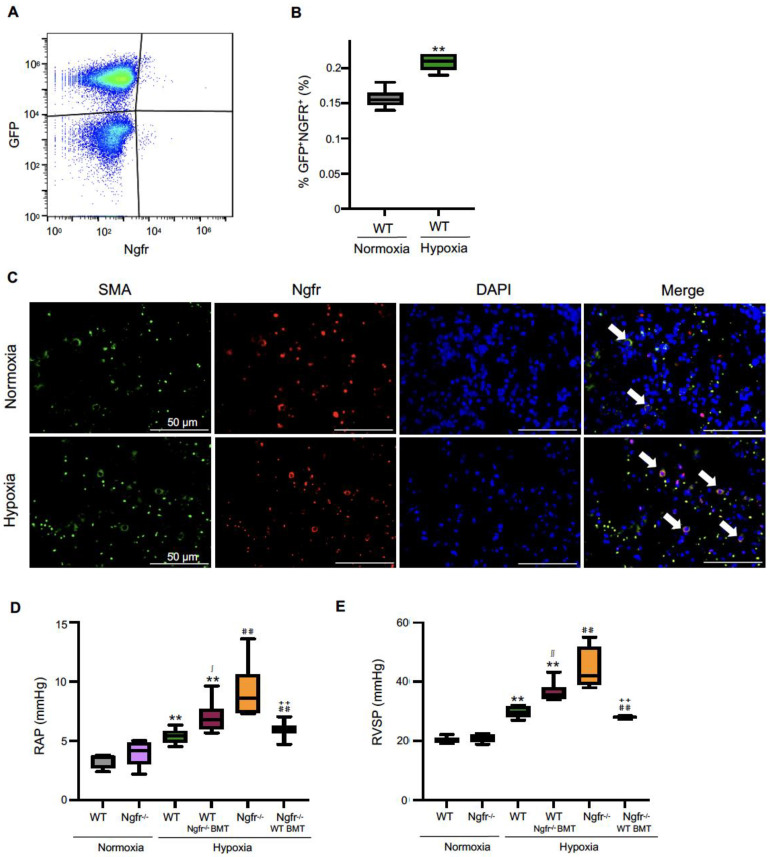
BM-specific depletion of Ngfr exacerbates hypoxia-induced pulmonary hypertension. (**A**) Representative flow cytometric figure (GFP+ Ngfr+ cells in lung tissue obtained from GFP-Tg BM-WT mice under hypoxic conditions). (**B**) The frequencies of GFP+ Ngfr+ double positive cells in lung tissue (% GFP+ Ngfr+) obtained from GFP-Tg BM-WT mice under normoxic and hypoxic conditions. (**C**) Immunofluorescence in the lung tissue of WT mice transplanted with GFP-Tg BM under normoxic and hypoxic conditions was used to assess the movement of circulating Ngfr-positive cells. GFP+Ngfr+ cells are indicated with white arrows (green: GFP, red: Ngfr, blue: DAPI, and Marge). RAP (**D**) and RVSP (**E**) in WT and Ngfr^−/−^ mice under normoxic and hypoxic conditions, and BM-specific Ngfr^−/−^ mice and BM replaced with WT under hypoxic conditions. BM, bone marrow; SSC, side scatter; FSC, forward scatter; DAPI, 4′,6-diamidino-2-phenylindole. GFP, green fluorescent protein; Ngfr, nerve growth factor receptor; WT, wild type; RAP, right arterial pressure; RVSP, right ventricular systolic pressure. Center lines show the medians, box limits indicate the 25th and 75th percentiles, whiskers extend 1.5× the interquartile range from the 25th and 75th percentiles. ** *p* < 0.01 versus WT Normoxia. ^∫^ *p* < 0.05 and ^∫∫^ *p* < 0.01 versus WT Hypoxia. ^##^ *p* < 0.01 versus Ngfr^−/−^ Normoxia. ^++^ *p* < 0.01 versus Ngfr^−/−^ Hypoxia.

**Figure 5 ijms-24-01868-f005:**
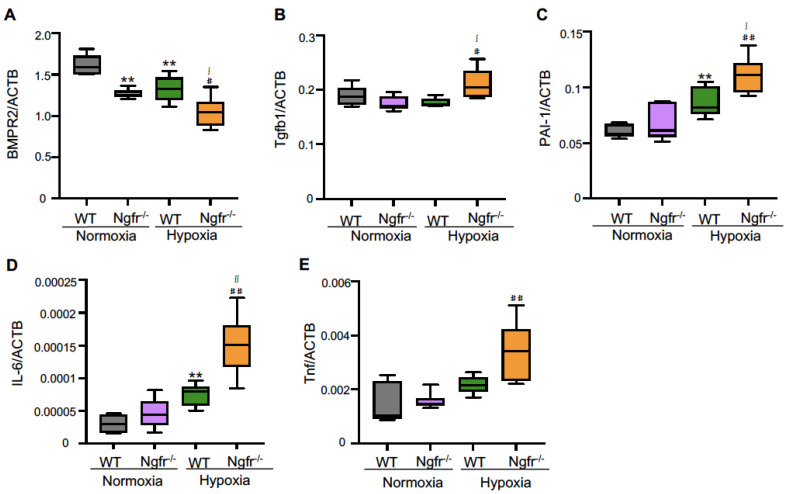
Gene expression profiles in lung tissue. Gene expression of BMPR2, Tgfb1, PAI-1, IL-6, and Tnf in WT and Ngfr^−/−^ mice under normoxic and hypoxic conditions. Gene expression of (**A**) BMPR2, (**B**) Tgf-b1, (**C**) PAI-1, (**D**) IL-6, and (**E**) Tnf in WT and Ngfr^−/−^ mice under normoxic and hypoxic conditions. BMPR2, bone morphogenetic protein receptor type 2; Tgf-b1, transforming growth factor-b1; PAI-1, plasminogen activator inhibitor-1; IL-6, interleukin-6; Tnf, tumor necrosis factor. Center lines show the medians, box limits indicate the 25th and 75th percentiles, whiskers extend 1.5× the interquartile range from the 25th and 75th percentiles. ** *p* < 0.01 versus WT Normoxia. ^#^
*p* < 0.05 and ^##^
*p* < 0.01 versus Ngfr^−/−^ Normoxia. ^∫^
*p* < 0.05 and ^∫∫^
*p* < 0.01 versus WT Hypoxia.

## Data Availability

Not applicable.

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
