# Peer review of "Important Role of Endogenous Nerve Growth Factor Receptor in the Pathogenesis of Hypoxia-Induced Pulmonary Hypertension in Mice"

_ijms, 2023, doi:10.3390/ijms24031868_

Round 1
Reviewer 1 Report
The Authors of the article entitled “Important Role of Endogenous Nerve Growth Factor Receptor in the Pathogenesis of Hypoxia-induced PH in Mice” have described an important role of Ngfr in hypoxia-induced PH. They have shown that the Ngfr gene deficiency enhances hypoxia-induced pulmonary vascular remodeling; and the gene expression of Ngfr and Ngfr positive cells were increased in hypoxic conditions in WT. Furthermore, hypoxia-induced PH was ameliorated in Ngfr-/- mice transplanted with WT BM cells. These are interesting results. However, Ngfr has been shown to have dual role (proliferative and apoptotic) in cancer. Freund-Michel et al (Am J Respir Crit Care Med. 2015;192(3):342-55.) have shown increased expression of Ngfr in experimental models of PH as well as in human PH. Interestingly, anti-NGF blocking antibodies prevented and reversed PH in rats, accompanied by significantly reducing vascular inflammation and remodeling. It would be worthwhile discussing theses differences.
Author Response
Reply)
The reviewer raised an important issue. We have added the following in the revised manuscript.
Discussion
Freund-Michel et al. reported that the administration of a neutralizing antibody to NGF, one low-affinity ligands of NGFR, suppressed hypoxia-induced PH in rats [14]. The authors showed that blockade of the NGF/NGFR axis suppressed the production of inflammatory cytokines in vascular smooth muscle cells, and proposed that NGF may mediate PH, These results are in contrast with our findings that the genetic deletion of NGFR promotes PH. This difference may be due to the fact that Freund-Michel et al. administered a neutralizing antibody for NGF to the hypoxia models, whereas we used a genetic deletion model of NGFR. Signaling through NGFR has different biological effects depending on where it can heterodimerize with other type of receptors (tropomyosine receptor kinase (Trk) A, B, and C; neurotensin receptor1, and 2; and sortilin) [15]. The co-expression of NGFR and Trks promotes cell survival, while the co-expression of NGFR and sortilin promotes cell death. The effects of the genetic deletion of NGFR on the activation of other types of receptors require further investigation.
(page 8, line 267 – page 9, line 284)
References
- Freund-Michel, V.; Cardoso Dos Santos, M.; Guignabert, C.; Montani, D.; Phan, C.; Coste, F.; Tu, L.; Dubois, M.; Girerd, B.; Courtois, A.; et al. Role of Nerve Growth Factor in Development and Persistence of Experimental Pulmonary Hypertension. Am J Respir Crit Care Med 2015, 192, 342-355, doi:10.1164/rccm.201410-1851OC.
- Bibel, M.; Barde, Y.A. Neurotrophins: key regulators of cell fate and cell shape in the vertebrate nervous system. Genes Dev 2000, 14, 2919-2937, doi:10.1101/gad.841400.
(page 14, line 521 – page 14, line 526)
Author Response
We thank the Reviewers for their careful review and constructive criticisms.

Round 2
